# Anti-HLA Class II Antibodies Correlate with C-Reactive Protein Levels in Patients with Rheumatoid Arthritis Associated with Interstitial Lung Disease

**DOI:** 10.3390/cells9030691

**Published:** 2020-03-11

**Authors:** Alma D. Del Angel-Pablo, Ivette Buendía-Roldán, Mayra Mejía, Gloria Pérez-Rubio, Karol J. Nava-Quiroz, Jorge Rojas-Serrano, Ramcés Falfán-Valencia

**Affiliations:** 1HLA Laboratory, Instituto Nacional de Enfermedades Respiratorias Ismael Cosío Villegas, 14080 Mexico City, Mexico; alyde_08@hotmail.com (A.D.D.A.-P.); glofos@yahoo.com.mx (G.P.-R.); krolnava@hotmail.com (K.J.N.-Q.); 2Sección de Estudios de Posgrado e Investigación Escuela Superior de Medicina, Instituto Politécnico Nacional, 11340 Mexico City, Mexico; 3Translational Research Laboratory on Aging and Pulmonary Fibrosis, Instituto Nacional de Enfermedades Respiratorias Ismael Cosio Villegas, 14080 Mexico City, Mexico; ivettebu@yahoo.com.mx; 4Interstitial Lung Disease and Rheumatology Unit, Instituto Nacional de Enfermedades Respiratorias Ismael Cosio Villegas, 14080 Mexico City, Mexico; medithmejia1965@gmail.com

**Keywords:** anti-HLA antibodies, panel-reactive antibodies, C-reactive protein, rheumatoid arthritis, interstitial lung disease, anti-CCP+, PRA

## Abstract

The pathogenesis of Rheumatoid Arthritis (RA) is not fully understood, probably influenced by genetic and environmental factors. Interstitial Lung Disease (ILD) is an extra-articular manifestation of RA, which contributes significantly to morbidity and mortality. The identification of anti-HLA antibodies has been useful in the transplantation field; however, its contribution to autoimmune diseases as RA has not been fully studied. We aimed to determine the presence of anti-HLA antibodies in RA patients with and without ILD and its possible association with clinical and biochemical markers. One-hundred and forty-seven RA patients, of which 65 had ILD (RA-ILD group), were included. Sera samples for Anti-HLA Class II LABScreen panel-reactive antibodies (PRA) were analyzed. In both groups, women predominated, and lung function was worse in patients with ILD. The anti-CCP+ (UI/mL) was higher in the RA group in comparison to RA-ILD (*p* < 0.001). Expositional risk factors (tobacco smoking and biomass-burning smoke) were higher in RA-ILD patients. PRA+ was identified in ~25% RA-ILD patients, while ~29% in the RA group. The CRP levels have a positive correlation with the percentage of reactivity (%PRA, *p* = 0.02, r^2^ = 0.60) in the RA-ILD group. In conclusion, anti-HLA antibodies correlate with C-reactive protein levels in RA patients with ILD.

## 1. Introduction

Rheumatoid Arthritis (RA) affects approximately 1% of the population worldwide. It is an inflammatory and autoimmune disease associated with joint destruction, and this can develop extra-articular complications [1,2]. Some extra-articular manifestations of RA include subcutaneous nodules, scleritis, and Interstitial Lung Disease (ILD), affecting the lung parenchyma [3,4], which contributes significantly to the morbidity and mortality of affected patients [5,6].

The pathogenesis of RA is not fully understood; the participation of genetic and environmental factors has been proposed. Among genetic factors are several alleles of the Human Leukocyte Antigen (HLA) system [7]. The HLA genomic region, located in chromosome 6 (6p21.3), is highly polymorphic, containing about 220 genes, many of which have immunoregulatory functions. The class I contains the classic genes HLA-A, -B, and -C, while HLA class II comprises the *HLA-DRB1*, -*DPB1* and -*DQB1* genes, encoding the β chain of proteins that are expressed as HLA antigens in specialized human cells [8]. The *HLA-DRB1*04* allele is the most replicated in association studies with the development of this disease [9].

The HLA has the task of recognizing the self from non-self-antigens; however, in RA, the immune system fails, and instead of reacting only against foreign antigens, it acts by attacking the joints, producing the characteristic phenomenon of autoimmunity [10]. This autoimmune reaction causes severe damage to other organs, which is caused by the generation of antibodies. The humoral response has been strongly related to anti-HLA Class I and Class II antibodies and is related to damage mechanisms linked to the endothelium and complement activation [11].

The generation of anti-HLA antibodies is due to different conditions, such as those that have a pathological origin (multiple blood transfusions, kidney, and other organ transplants) [12]. Some studies have found that anti-HLA antibodies, a product of blood transfusion, increases the risk of lung injury and is one of the causes of transfusion-related mortality [13]; moreover, they can also be considered as a product of common physiological conditions (maternal-fetal alloimmunization in multiparous women) [14,15], resulting in microchimerism, which is the persistence of foreign cells in an individual [16]. Recently, some studies have described microchimerism as a possible genetic contribution to autoimmune diseases such as systemic sclerosis, where one of the causes could be exposure of the maternal immune system to allogeneic antigens on fetal microchimeric cells, which would produce cytotoxic antibodies [17,18]. Antibodies against HLA antigens are frequently detected in the sera of women and increase during parity [13,19]. Pregnancy can also have important implications for an autoimmune disease such as systemic lupus erythematosus (SLE) and RA, with women having a much higher risk of a RA diagnosis during and following their childbearing years [19,20].

The RA diagnosis is based on clinical manifestations, radiological, and laboratory parameters, including anti-cyclic citrullinated peptide antibodies (anti-CCP) and rheumatoid factor (RF) detection. Inflammatory markers such as C-reactive protein (CRP) and erythrocyte sedimentation rate (ESR) can serve to follow the illness progress [21]. CRP is an indicator parameter of inflammatory activity in the acute phase. It is employed to control the response to treatment in RA patients [22]; however, it is not specific since it is affected by numerous factors [23,24,25].

The anti-HLA antibodies detection is accomplished in the organ transplants’ field; however, its participation in autoimmune diseases has not been thoroughly studied and could help to understand the mechanism of disease and development of complications. Therefore, we aimed to determine the presence of anti-HLA antibodies in patients with rheumatoid arthritis with and without ILD and its possible association with clinical and biochemical markers.

## 2. Materials and Methods

### 2.1. Study Population

A case-control study, including 147 subjects diagnosed with RA, was designed; Sixty-five patients having ILD, were included. All of them recruited from the Interstitial Lung Disease and Rheumatology Unit at the Mexican Instituto Nacional de Enfermedades Respiratorias Ismael Cosio Villegas (INER) in Mexico City. All patients with RA met the American College of Rheumatology (ACR/EULAR 2010) [21], and Thoracic Society/European Respiratory Society (ATS/ERS) criteria [26]. The disease activity was evaluated by Simplified Disease Activity Index (SDAI) that include C-reactive protein (mg/dL) as well as the patient’s global assessment of health on a 10 cm visual analogue scale (PrGA = provider global assessment of disease activity and PtGA = patient global assessment of disease activity) [27]. The RA-ILD diagnosis was assessed by the presence of typical features in the lung by high-resolution chest tomography. Clinical and radiological data from medical records were obtained. The pulmonary function test (spirometry was performed to evaluate forced expiratory volume in the first second (FEV_1_), forced vital capacity (FVC), and the FEV_1_/FVC ratio. The values were expressed as a percentage and an FVC < 80% was defined as a restrictive pattern. Levels above 30 UI/mL of anti-CCP and 20 UI/mL of RF, respectively, were regarded as positive.

### 2.2. Ethics Statement

The Institutional Committees for Research, Biosecurity, and Ethics in Research of the Instituto Nacional de Enfermedades Respiratorias Ismael Cosío Villegas (INER) approved this study (approbation codes: B20-15 and C08-15). All participants authorized and signed the corresponding informed consent.

### 2.3. Anti-HLA Antibodies Detection

Sera samples from peripheral blood from 147 patients through venipuncture were obtained. Samples processing began with centrifugation for 5 min at 4500 rpm to separate the serum of the whole blood. Anti-HLA Class II LABScreen panel-reactive antibodies (PRA) detection was done according to manufacturers’ indications. The PRA test is based on a microsphere-based multiple fluorescence immunoassay using the Luminex platform, used to detect anti-HLA antibodies in response to organ transplantation [28,29]. Briefly, samples were analyzed for Anti-HLA Class II LABScreen PRA (One Lambda, West Hills, CA, USA). 20 µL of serum was incubated with the beads for 30 min at room temperature in the darkness. The beads were washed three times with 1X wash buffer, and 100 µL goat anti-human IgG conjugated with R-Phycoerythrin (PE), and diluted 1:100 in 1X wash buffer, was added. After the 30 min incubation step at room temperature in the darkness, finally, buffer PBS 1X was added. Samples were analyzed using the LABScan™ 100 (Luminex^®^ 100, Austin, TX, USA) system. The data obtained were analyzed by using HLA Fusion™ 4.0 Software (One Lambda, West Hills, CA, USA) [30].

### 2.4. Statistical Analysis

Categorical variables were described with numbers and percentages, and continuous variables with median and minimum-maximum (Min-Max) values. The differences in the characteristics of the patients were analyzed using the Mann–Whitney U test, the χ^2^ test using 2 × 2 contingency tables, and correlation analyses using the Pearson correlation test. Statistical significance was defined as *p* < 0.05. The statistical analyses were performed using R Studio v. 3.6.1. (R Core Team, Vienna, Austria) [31] and Epi-Info v.7.2.2.6 (CDC, Atlanta, GA, USA) [32].

## 3. Results

The current study included 147 patients divided into two groups: one with a diagnosis of RA-ILD (*n* = 65) and another only with RA (*n* = 82). After the PRA test was performed, the number of patients who tested positive were: RA-ILD (*n* = 16) and RA (*n* = 24). See Figure 1.

The percentage of women in the study groups exceeded 50.0%, being the RA-ILD group, where the lowest percentage (53.0%) was found compared to the RA group with 81.0 % (*p* = 0.002). In the age of diagnosis of RA, again, the RA-ILD group was the group that distinguished itself from the rest by including the older participants compared to the RA group (*p* = 0.001). As for data of the SDAI, patients with RA had higher articular disease activity in comparison with the patients who had a diagnosis of RA-ILD; however, there were no statistically significant differences between groups of patients: RA-ILD (SDAI = 28.10) vs. RA (SDAI = 32.25). See Table 1.

Regarding anti-CCP antibodies, in the RA group, 95.2% of the patients were positive compared to the RA-ILD group where 83.1% were positive (*p* > 0.05); however, in the titers’ (UI/mL) comparison, the RA group showed a marked increase concerning the RA-ILD group (*p* < 0.001). In the opposite way, with the RF, the highest levels were found in the group of patients who had RA-ILD, where the percentage of positive patients reached > 90.0% (information available in Appendix A). Regarding the CRP values, the RA-ILD group had higher levels compared to the group patients diagnosed only with RA (1.56 mg/dL vs. 1.40 mg/dL). Regarding the ESR, the patients with ILD (32 mm/h) presented higher values than patients with only RA (26 mm/h), however, for both parameters there were no statistically significant differences found (Table 1).

In the lung function analysis, the RA-ILD group had a lower FEV_1_ with a value of 66% compared to the RA group (81%). Regarding lung function values, a significant difference was detected between groups, as revealed by the values of FEV_1_, FVC, and the FEV_1_/FVC ratio (*p* <0.001). See Appendix A. 

Concerning tobacco smoking and biomass-burning smoke exposition, the highest exposure rates were in the RA-ILD group. The exposure of organic and inorganic compounds was higher in patients with ILD. Table 1 shows the characteristics of the groups.

In this study, all the patients (with or without ILD) received disease-modifying antirheumatic drugs (DMARD) as a first-line treatment, the percentage treated with Methotrexate was higher than 90.0%. Additionaly, Leflunomide, in the RA-ILD group had a higher percentage (42.9%) compared to the RA group (28.2%). For the proportion of patients who had Sulfasalazine therapy, there was a significant difference (*p* < 0.001) in RA-ILD (19.1%) vs. RA (65.4%). The least number of patients received medication with biological DMARDs (bDMARDs); for the group of patients with RA-ILD, the proportion was ~ 10.0%, and for the RA group, it was < 2.0%. Both groups were treated with immunosuppressive therapy, and in the RA-ILD group almost 70.0% were treated with prednisone compared to the ~ 40.0% in the RA group (*p* < 0.001). Only the patients who had ILD were treated with Azathioprine. Most patients have combined therapy with two to three DMARDs (or combined with a biological agent). In patients who have biological therapy, approximately 8.0%, have treatment with B-cell targeted therapy (rituximab) and only one patient has anti-TNF treatment (adalimumab) (See Table 1).

### 3.1. Clinical features of the ILD patients

As for lung disease pattern, this was heterogeneous; nevertheless, the Nonspecific Interstitial Pneumonia (NSIP) pattern was the most prevalent with 73.33%. Additionally, Usual Interstitial Pneumonia (UIP = 16.67%), Cryptogenic Organizing Pneumonia (COP = 5.0%) and Lymphocytic Interstitial Pneumonia (LIP = 3.33%) were patterns less frequent in patients with ILD (Appendix A).

The patients with NSIP had lower titers of anti-CCP+ (176.2 IU/mL) in comparison to the UIP (201 IU/mL), COP (175 IU/mL) and LIP (268.3 IU/mL) patterns. The patients with NSIP (1.84 mg/dL) had higher values compared to UIP (0.42 mg/dL). While the patients with an NSIP pattern had a lower percentage of lung function (FEV_1_ = 63.5%) in comparison to the UIP, COP, and LIP patterns with an FEV_1_ value > 80%.

### 3.2. Panel-Reactive Antibodies (PRA) Analysis

We evaluated the frequency of anti-HLA antibodies positive-patients, finding 24 in the RA group and 16 for RA-ILD; positive subjects were mostly women who had previous pregnancies and a male patient for the RA-ILD group, who had a history of blood transfusions. The group with a higher percentage of reactivity was RA (23%), while the RA-ILD group was 20%; the most frequent anti-HLA for both groups were DR4, DR8, DR12, and DQ9 antigens.

Regarding lung function, the RA-ILD group had a lower lung function (61%) compared to the RA group (94%), shown in Appendix A. The NSIP pattern appeared in ~74.0% of the RA-ILD group, followed by UIP and LIP (~13% each one). The PRA reactivity between patterns was as follow: the NSIP pattern group reached 20.0% for PRA, while only two patients had UIP (~10.0% PRA) and LIP (56.0% PRA). This can be reviewed in Appendix A

There were no statistical differences in the RF, CRP, and neither ESR levels in the study groups. The RA patients had higher anti-CCP+ levels in comparison with RA-ILD patients; this difference was statistically significant (*p* = 0.032), the corresponding box-plot is in Appendix A. In general, the values maintained the same behavior in terms of the full study population. However, patients with RA-ILD have greater exposure to environmental pollutants (smoking, biomass-burning’ smoke, organic, and inorganic compounds); these differences were not statistically significant. See Table 2.

Once performing correlations between clinical and biological markers of the disease with pulmonary function variables, no statistically significant values were found; however, we identified a relationship between anti-HLA antibodies with the CRP levels, and we found a positive correlation with the percentage of reactivity, (*p* = 0.02, r^2^ = 0.60) in the RA-ILD group. See Figure 2.

## 4. Discussion

Interstitial Lung Disease (ILD) is a major complication in Rheumatoid Arthritis (RA) patients [33], where clinical manifestations may differ among patients with and without ILD. In the current study, we were able to identify that the ILD diagnostic in patients with RA was made around 60 years old, additionally, in the RA-ILD group, the proportion of men was higher than in the RA group; this corresponded with that reported in other studies conducted in Caucasian and Asian patients [5,6]. In our study, the most frequent tomographic patterns were NSIP and UIP. The NSIP pattern can have an inflammatory predominance over fibrosis. In other studies, the patterns of progression most frequent were UIP, followed by NSIP and OP; in previous studies, these patterns carry a higher mortality rate than UIP [34]. FEV_1_ was generally lowest in patients with ILD compared with RA patients without pulmonary affection. Low FEV_1_ is generally considered one of the most sensitive parameter for early detection of ILD [35].

Among the environmental risk factors for RA, smoking is the most critical predisposing it [36,37]. Tobacco smoking leads to a higher expression of the PAD2 enzyme in the lung [38]. The frequency de RA-ILD was higher in men than in women, and this may be related to a higher incidence of tobacco smoking in men than in women [39,40]. These descriptions in other populations match with our results.

Additionally, RA patients with ILD had higher RF titers in comparison with the RA patients; this was in agreement with previous studies [6]. The anti-CCP+ levels were low in RA-ILD patients; the latter has been considered a highly specific RA diagnostic marker [41]. Interestingly, we found that 100% of our RA-ILD patients were anti-CCP+ in PRA positive patients. In other studies, it has been reported that anti-CCP positivity correlates with the presence of ILD in RA and higher titers of anti-CCP+ may be associated with more severe ILD [6,42,43,44], nevertheless, in our study lower titers of anti-CCP+ were found in the RA-ILD patients in comparison to patients with RA without ILD. This could be the result of better disease control in patients with ILD compared to that of RA patients; further, ILD patients receive conventional DMARDs and in some cases a biological agent. Several studies have described the involvement of anti-TNF in combination with DMARDs in modulating anti-CCP levels [45,46], although researchers describe that stricter studies should be conducted to elucidate this response. Other biological agents in which a better response has been seen in the decrease of anti-CCP levels are with B-cell targeted therapy (rituximab) [47,48]. In the current study, only five RA-ILD patients received this therapy.

The therapy in RA-ILD patients is based according to the underlying disease (RA). Several studies have described therapy with DMARDs and corticosteroids (mainly prednisone) that help modulate the CRP and ESR levels, and are considered as monitors for successful treatment [49,50]. Azathioprine is used as an alternative therapy to corticosteroids and a complementary agent mainly to infliximab and adalimumab, however, it is used in moderate and severe cases [51]. Immunosuppressive agents (prednisone and azathioprine) help to modulate the main treatment dose and to avoid adverse events in patients.

We have identified, for the first time, anti-HLA antibodies in RA-ILD and RA patients; in other studies HLA cytotoxic antibodies were detected in transfused patients; a positive correlation between the transfused load and the frequency of alloimmunization against HLA antigens was also found [52]. With regard to anti-HLA Class II antibodies, patients who received a heart transplant developed IgG antibodies compared to control subjects [53]. Studies of anti-HLA Class I and II antibodies that participate in hyperacute rejection in pulmonary allograft have been developed, acquiring bronchiolitis obliterans syndrome [53,54].

Several studies, in the transplantation field, have shown that anti-HLA antibody profiles correlate to the rate of allograft rejection [12,55], however, these studies have not explored the role it performs in autoimmune diseases. Between the potential sources of anti-HLA antibodies are blood transfusions, since a probable complication is transfusion-related acute lung injury (TRALI), caused by the anti-HLA Class I and Class II antibodies pre-existents in the plasma donors, where the infused blood stimulates pulmonary endothelial cells and therefore increased capillary permeability [56]. However, no blood transfusions were reported among the included PRA positive patients, except for a male patient in the RA-ILD group. The rest of the positive patients were female with pregnancy antecedent, in this regard, previous studies have revealed that fetal microchimerism influences the alloimmunization [13].

To the best of our knowledge, there are no studies that have linked the biochemical (laboratory testing) parameters with the percentage of reactivity (PRA%) in RA-ILD patients. In 2012 Furukawa and collaborators performed a study where they analyzed the profile of antibodies anti-HLA (Class I, Class II and MICA proteins), finding anti-HLA antibodies (MICA/Class I) in collagen vascular disease-associated ILD [57]. Another study found that levels and frequencies of anti-HLA antibodies were higher among women and males with SLE and RA [58]. Our results in the RA group showed a higher percentage of reactivity.

In this study, we tested the influence of the anti-HLA antibodies with different clinical and biological RA markers, identifying a correlation with CRP levels. CRP is a protein produced by hepatocytes; this biomolecule increases rapidly as a response to inflammation-mediated by IL-6, IL-1β, and TNF-α [59]. Different studies have shown the usage of CRP as a predictor of cardiovascular diseases [60], however, there are no studies that correlate CRP levels with the severity of RA and extra-articular complications due to anti-HLA antibodies.

High CRP levels correlate to the progression of the RA disease [61]; in our study, we found higher titers in the RA-ILD group compared to the group of RA patients. Since CRP plays a pathogenic role in the inflammation observed in RA, CRP may be associated with a severe RA manifestation, for example, ILD. In patients with RA-ILD higher CRP titers were found compared to patients with RA but without ILD [6]. In another study conducted in ILD patients with non-pulmonary surgery, they found high CRP levels; authors consider it a predictor of acute exacerbation [62]. However, the pathogenic mechanisms of CRP and anti-HLA antibodies in both diseases have not been thoroughly investigated. In our study, we found a positive correlation between CRP levels and the percentage of panel-reactive antibodies (PRA) in the RA-ILD group. The relation between these two factors is in the activation of the complement system, where CRP has an interaction with the C1q complex, which results in the activation of the classical complement cascade [63]. In our results, the correlation was found in the group of ILD patients, and this could be due to the ILD, which is an extra-articular complication of RA. Low doses of CRP could modulate autoimmune diseases and prevent the generation of autoantibodies [64]. Our findings could help to elucidate the possible implication of anti-HLA antibodies/CRP in patients with RA with an extra-articular manifestation such as ILD.

This study was not free of limitations. Possibly the most important was that we were not able to identify specific HLA-antigens which the anti-HLA antibodies were addressing. The panel-reactive antibodies is not specific for the detection of the single antigens. This assay helps us to detect the percentage of reactivity (%PRA) and identifies possible groups of antigens (such as DR4, DR8, and DQ9) that the subjects harbor. The single antigen detection requires a second commercial kit to detect specific antigens that occur in patients who have a higher percentage of reactivity.

Furthermore, this was a retrospective study and can have memory bias (in expositional variables referred by participants, as well as blood transfusions); RA-ILD prevalence is low and probably underdiagnosed, and so, the sample size is small; Among strengths, this is the first paper evaluating the anti-HLA antibodies using the PRA test in an autoimmune disorder with interstitial lung disease.

## 5. Conclusions

There is a correlation between CRP levels with the panel-reactive antibodies (PRA) percentage of the HLA Class II antigens. The RA patients with ILD had higher CRP levels compared to patients who had only RA, which could indicate that there is a higher degree of inflammation. CRP levels can be modified by treatment; however, patients who have RA-ILD have higher levels compared to patients who only have RA. The identification of anti-HLA antibodies in patients with RA can have an impact on the severity of the disease and potentially with the response to the treatment of interstitial lung diseases.

## Figures and Tables

**Figure 1 cells-09-00691-f001:**
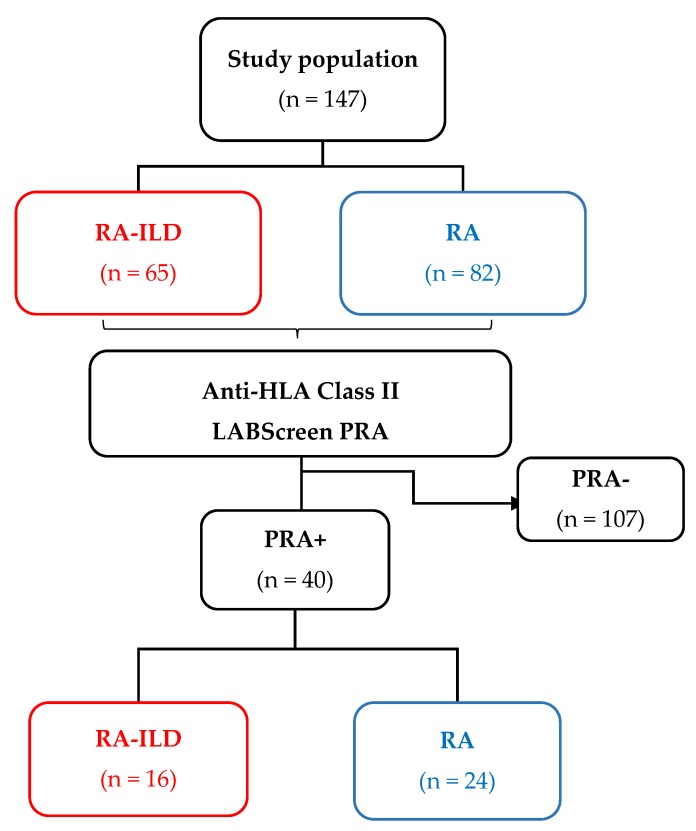
Sera panel-reactive antibodies (PRA) analysis included samples from the 147 Rheumatoid Arthritis (RA) patients (with or without Interstitial Lung Disease (ILD)) for the LABScreen system.

**Figure 2 cells-09-00691-f002:**
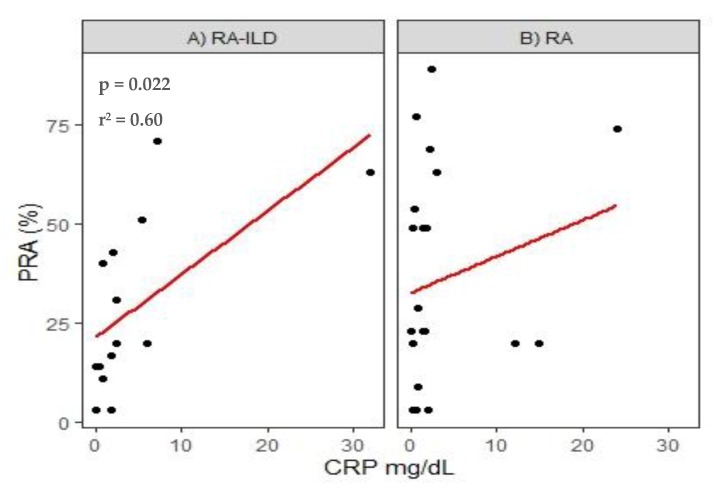
Correlation of CRP levels with PRA (%) in rheumatoid arthritis (RA) patients with or without ILD.

**Table 1 cells-09-00691-t001:** Demographics and clinical characteristics of the included subjects.

Variable	RA-ILD *n* = 65	RA *n* = 82	*p*-value
Female (%)	53 (81.54)	81 (98.78)	0.002
Age	61 (37–85)	53.50 (25–80)	<0.001
Age at RA diagnosis	53 (23–85)	45.50 (18–75)	<0.001
Age at ILD diagnosis	59 (37–85)	NA	
FEV_1_* (%)	66 (18–145)	98 (48–134)	<0.001
FVC* (%)	67 (26–147)	97 (54–128)	<0.001
FEV_1_/FVC* (%)	82 (30–110)	99 (58–117)	<0.001
anti-CCP+, *n* (%)	49 (83.05)	59 (95.16)	
anti-CCP+ (UI/mL)	190.91 (44.77–378)	322.6 (42.5–531)	<0.001
RF+, *n* (%)	56 (94.92)	61 (93.85)	
RF+ (UI/mL)	291 (20.20–4550)	182 (20.40–2100)	
CRP (mg/dL)	1.56 (0.01–32)	1.40 (0.05–24)	
ESR (mm/h)	32 (1–45)	26 (2–110)	
PRA+, *n* (%)	16 (24.62)	24 (29.27)	
SDAI	28.10 (1–76.01)	32.25 (1.31–100.0)	
*Exposure*			
Tobacco smoking, *n* (%)	22 (37.93)	19 (31.67)	0.003
Tobacco index	5.28 (0.10–78)	2 (0.20–10.50)	0.04
Biomass-burning exposition, *n* (%)	23 (43.40)	16 (30.77)	
Biomass-burning exposition index	79 (14–638)	34 (3–414)	0.014
Exp-Org, *n* (%)	33 (52.38)	36 (48.65)	
Exp-Inor, *n* (%)	21 (33.87)	24 (31.17)	
*Treatment*			
Methotrexate, *n* (%)	59 (93.65)	76 (97.44)	
Leflunomide, *n* (%)	27 (42.86)	22 (28.21)	
Sulfasalazine, *n* (%)	12 (19.05)	51 (65.38)	<0.001
Cloroquine/ Hydroxychloroquine, *n* (%)	22 (34.92)	54 (69.23)	<0.001
bDMARDs, *n* (%)	6 (9.52)	1 (1.28)	
Azathioprine, *n* (%)	8 (12.70)	0	
Prednisone, *n* (%)	44 (69.84)	31 (39.74)	0.001

Median (Min-Max). Mann–Whitney U test and χ^2^ test using 2 × 2 contingency tables were employed to calculate *p*-values. Only statistically significant values are shown (*p* <0.05). Six-patients with rituximab (5 in RA-ILD, 1 in RA), and one patient with adalimumab (RA-ILD). RA: Rheumatoid Arthritis; ILD-RA: RA with Interstitial Lung Disease; *Pre-Bronchodilator; FEV_1:_ forced expiratory volume in the first second; FVC: forced vital capacity; FEV_1_/FVC: forced expiratory volume in the first second/forced vital capacity ratio; CRP: C-reactive protein; anti-CCP+: anti-cyclic citrullinated peptide antibody; RF: rheumatoid factor; Exp-Org: organic exposure; Exp-Inor: inorganic exposure; PRA: panel-reactive antibodies, bDMARDs: Biological DMARD.

**Table 2 cells-09-00691-t002:** Demographic and clinical characteristics of PRA positive patients.

Variable	RA-ILD *n* = 16	RA *n* = 24	*p*-Value
Female (%)	15 (93.75)	24 (100)	
Age at RA diagnosis	52 (32–85)	46.5 (23–69)	
FEV_1_* (%)	61 (25–145)	94 (62–129)	<0.001
FVC* (%)	62 (26–147)	97 (77–126)	<0.001
FEV_1_/FVC* (%)	82.25 (49–93.30)	94 (58–117)	<0.004
anti-CCP+, *n* (%)	16 (100.0)	17 (89.47)	
anti-CCP+ (UI/mL)	195.96 (56.5–345.6)	244.25 (63.2–501)	0.032
FR+, *n* (%)	16 (100)	16 (94.12)	
FR+ (UI/mL)	242 (34.9–2970)	220.6 (20.4–1647.7)	
CRP (mg/dL)	1.94 (0.01–32	1.38 (0.05–24)	
ESR (mm/h)	35.5 (1.49–40)	29 (2–110)	
Tobacco smoking, *n* (%)	5 (33.33)	3 (16.67)	
Tobacco index	7.3 (0.10–23.5)	2.4 (0.80–5.0)	
Biomass-burning exposition, *n* (%)	8 (61.54)	4 (23.53)	
Biomass-burning exposition index	99.5 (25–400)	32.5 (14–414)	
Exp-Org, *n* (%)	7 (46.67)	10 (41.67)	
Exp-Inor, *n* (%)	4 (28.57)	5 (21.74)	
PRA (%)	20 (3–71)	23 (3–89)	

Median (Min-Max). Mann–Whitney U test and χ^2^ test using 2 × 2 contingency tables were employed to calculate *p*-values. Only statistically significant values are shown (*p* <0.05). RA: Rheumatoid Arthritis; ILD-RA: RA with Interstitial Lung Disease; *Pre-Bronchodilator; FEV_1:_ forced expiratory volume in the first second; FVC: forced vital capacity; FEV_1_/FVC: forced expiratory volume in the first second/forced vital capacity ratio; CRP: C-reactive protein; anti-CCP+: anti-cyclic citrullinated peptide antibody; RF: rheumatoid factor; Exp-Org: organic exposure; Exp-Inor: inorganic exposure; PRA: panel-reactive antibodies.

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
