# Peer review of "Anti-HLA Class II Antibodies Correlate with C-Reactive Protein Levels in Patients with Rheumatoid Arthritis Associated with Interstitial Lung Disease"

_cells, 2020, doi:10.3390/cells9030691_

Round 1

Reviewer 1 Report

No!

Author Response

Thank you!

Reviewer 2 Report

This revised manuscript is well writen. I don't have any additional comments. 

Author Response

Thank you for your kindly comments.

Reviewer 3 Report

See attached pdf file

Author Response

Major points:

Authors should avoid talking about test for assessing anti-HLA antibodies in introduction, but in my opinion, it would be more suitable mention it in methods. Introduction should be streamlined for a better reader comprehension. It is poor clear the role of HLA in Rheumatoid arthritis. I suggest improving this topic in the introduction.

Thank you very much for the suggestions; all were taken into account. In the current version, we have moved the anti-HLA antibodies testing to the methods section. Also, the introduction section has been improved to accomplish your suggestion.

New classification criteria currently are employed in clinical trial. Please, replace older criteria with more recent one, and authors verify that patients included in the study meet the new criteria. [Aletaha D, Neogi T, Silman AJ, Funovits J, Felson DT, Bingham CO, 3rd, et al. 2010 Rheumatoid arthritis classification criteria: an American College of Rheumatology/European LeagueAgainst Rheumatism collaborative initiative. Arthritis Rheum. 2010;62:2569–81].

Thank you for your advice; unfortunately, this was a mistake in the reference cited, the diagnosis was based on the updated criteria of the ACR/EULAR; however, we were wrong to put the reference. The right reference was replaced.

Authors should state in methodology section if they used CRP or ESR for assessing disease activity through DAS-28. Of note, DAS28-CRP and DAS28-ESR cut-offs for disease activity in rheumatoid arthritis are not interchangeable. In this regard, Fleischmann et al. have recently suggested that use of the DAS28-ESR cut-off to assess disease activity may underestimate the number of patients with high disease activity.

Reviewer's comment is right, in the current version, we have amended this sentence, describing that the CRP parameter was used to calculate DAS28.

DAS28-ESR, CDAI and SDAI are the main tools to evaluate disease activity in RA. However, they weight their individual components differently, sometimes causing discordant assessments of RA disease activity. Based on these findings, authors should consider also CDAI and SDAI for assessing disease activity.

This is a very important comment; unfortunately, we do not have the visual analogue scale necessary to calculate CDAI or SDAI indexes.

In my opinion, treatments, especially biologics, should be mentioned, preferably in a table. Indeed, it is well known as treatments may affect inflammatory parameters such as ESR and CRP. Therefore, among biologics, authors should specify which biologics they gave to the patients. In this regard, it is worth mentioning how inflammatory parameters, appreciably drop-off with anti IL-6 treatments, thus it is clear that results may potentially be affected by treatment.

Thank you for your comment, now we have included the treatment in the table of clinical variables. None of the patients reported having T-cell target therapy (IL-6).

Literature evidence highlights that anti-TNF agents might be associated with interstitial lung disease adverse events in RA and can induce more severe pulmonary symptoms and even result in death. Did the authors rule out that ILD in some of their patients could be due to adverse event from anti-TNF agents? For this reason, it is important to state in a table, treatments that were gave to the patients.

Thank you for your kindly advice, the patients who received biological treatment, most of these patients received treatment with B-cell target therapy (Rituximab), and only one patient received anti-TNF treatment (Adalimumab), now this stated in the results section.

In my opinion, there is another major limitation of the study: at the time of samples collection, all of the patients were already taking immunosuppressive agents, especially biologics, which might have affected inflammatory parameters such as CRP. I suggest stating this important bias in this study.

Thank you very much for the observation; almost 8% of RA-ILD patients received biological therapy. The rest of the patients were treated with non-biological DMARDs. Now, this is described in the corresponding sections.

Finally, two native speakers have reviewed the whole manuscript for grammar and typos.

Round 2

Reviewer 3 Report

Please see attached pdf file

Author Response

We appreciate the further comments of the Reviewer that allow us to clarify some points in the manuscript.

  1. The CDAI or Clinical Disease Activity Index and the SDAI or Simplified Disease Activity

Index are inspired by the « DAS » score family for Rheumatoid Arthritis, comprising namely DAS28-ESR and DAS28-CRP.

They are very useful to make an objective, reproducible and comparable assessment of the rheumatoid arthritis activity.

SDAI takes into account the following items: TJC28: The number of tender joints (0-28). SJC28: The number of swollen joints (0-28). CRP: The C-Reactive Protein concentration (in mg/dl, between 0 and 10).

PaGH: The patient global health assessment (from 0=best to 10=worst).

PrGH: The care provider global health assessment (from 0=best to 10=worst).

On the contrary CDAI takes into account all the above-mentioned items without CRP.

Currently there are several online calculators that allow us to assess SDAI and CDAI simple enough. On this basis the authors could recover some items from DAS28-CRP already evaluated to calculate their patients' SDAI and CDAI.

Thanks very much for the suggestion; the CDAI/SDAI indexes were calculated with the help of our rheumatology specialist' team. We decided to use the SDAI score since of its simplified assessment, and this score also includes de CRP as an acute inflammation reactant. Some studies have compared this score versus DAS28 and demonstrated that it might be useful. The current version has some descriptive paragraphs, including this information.

  1. Authors did not reply correctly to my comment:

Leaving out the biologicals (only 8% of the patients took these drugs) at the time of samples collection, the majority of patients were already taking immunosuppressive agents including methotrexate, leflunomide and so on, besides glucocorticoids (prednisone) which might have affected inflammatory parameters such as CRP. I suggest stating this critical bias in the conclusions of study.

Thanks for the observation, it was taken into account and described in the manuscript about the immunosuppressive therapy status.

This manuscript is a resubmission of an earlier submission. The following is a list of the peer review reports and author responses from that submission.

Round 1

Reviewer 1 Report

Authors demonstrated anti-HLA antibodies correlate with C-reactive protein levels in RA patients with ILD. The study was well designed. However, RA disease activity or ILD activity was less addressed.

Reviewer 2 Report

Dear Author!

 I was asked to review the manuscript ": Anti-HLA antibodies correlate with C-reactive protein levels in patients with Interstitial Lung Disease and Rheumatoid Arthritis".

I have to say that the introduction, methods and results as well as discussion session were clear. My main concern is that we actually could not learn too much from this study, partially because of the small patients' number but mainly because of the absence of patients' characteristics even in the light of lung changes (pattern of ILD for example) or joint diseases activity (may be CRP comes from active RA?), treatments or response to treatments. Actually, there is no clear what will teach us the presence of anti HLA antibodies regarding the RA or RA-ILD.

Best regards

Reviewer

Reviewer 3 Report

The authors investigated the rate of anti-HLA-class II autoantibodies by PRA technology in patients with RA and those complicated with interstitial lung disease (RA-ILD). Then, they tried to correlate % PRA with other Lab in these patients. The authors concluded that %PRA positively correlates with serum CRP levels of RA-ILD. The following comments and queries are raised for authors’ references.

The title of the manuscript seems inappropriate. It should be “Anti-HLA class II antibodies correlate with C-reactive protein levels in patients with rheumatoid arthritis associated 9or complicated) with interstitial lung disease”. The rationale of the present study is vague. The authors only stated that the participation of anti-HLA antibodies in autoimmune diseases has not been studied in literature in P.2 line 69. However, Furukawa et al. (Ref. 46) have reported anti-HLA class I-related chain A (MICA) as markers of ILD after extensively survey of HLA class I, class II, and MICA. Are these anti-HLA-class II antibodies detected by authors derived from alloantigen-induced or polyclonal B cell activation? In general, HLA-class I antigens are presented in all nucleated cells. In contrast, HLA-class II antigens are restricted in the surface of antigen-presenting cells. It is deduced that the chronic inflammation in ILD would be majorly caused by anti-HLA class I rather than class-II antibodies or immune complexes. The authors have to explain why only detect anti-HLA-class II autoantibodies. Table 1: Why the percentage and titer of anti-CCP in RA-ILD is conversely decreased in RA-ILD although tobacco smoking in this group is greater than RA group. Table 2: The PRA (%) in RA-ILD is less than RA group. This may indicate PRA is not absolutely related to the occurrence of ILD in RA patients. However, the authors concluded the production of anti-HLA class II PRA is positively related to CRP levels in RA-ILD. The authors have to explain more in the “Discussion” part. The authors are requested to explain why anti-HLA antibodies are produced in some particular RA patients? What is the pathological role of the autoantibodies in inducing ILD? Why these particular antibodies correlate with CRP levels in RA-ILD? Can these anti-HLA-class II PRA-detected antibodies against self-class II antigens in patients with RA? Or, these anti-HLA-antibodies are only cross-reactive with other somatic antigens rather than HLA antigens?

Reviewer 4 Report

The manuscript  needs  extensive  editing  of English language  and  the  study  itself  needs  more  stringent criteria  for  classifying the  patients  into  RA ILD   and  only  RA. 

the topic of  RA-ILD is  important  however,  I  believe the study  does not follow  a  well planned  design.